# Evolution of Shock Waves during Muzzle Jet Impinging Moving Bodies under Different Constrained Boundaries

**Zijie Li * and Hao Wang**

School of Energy and Power Engineering, Nanjing University of Science and Technology, Nanjing 210016, China; wanghao1960@126.com
* Correspondence: lizijie@njust.edu.cn

**Abstract:** A recently developed launching device called the gun–track launch system is affected by its constrained track, such that the form of the muzzle jet changes from the state of free development in the entire space to a constrained state, where this lends unique characteristics of development to its flow field. In this study, the authors establish the corresponding model for numerical simulations based on the dynamic mesh method. We also considered a model of simulation of the muzzle jet with an "infinitely" constrained track to analyze its performance under real launch conditions to explore the mechanism of development and the disturbance-induced propagation of the shock wave when the muzzle jet impinges on moving bodies. The results showed that the muzzle jet exhibited a circumferential asymmetric shape that tilted toward the area above the muzzle and generated transverse air flow that led to the generation of a vortex on it. Because the muzzle was close to the ground, the jet was reflected by it to enhance the development and evolution of the shock waves and vortices and to aggravate the rate of distortion and asymmetry of the jet. The wave reflected from the ground was emitted once again when it encountered the infinitely constrained track. No local low-pressure area or a prominent vortex was observed after multiple reflections. Because the track in the test model was short, the waves reflected by the ground were not blocked, and vortices were formed in the area above the ground. Significant differences in the changes in pressure were also observed at key points in the domain. The results of a comparative analysis showed that the infinitely constrained track increased the Mach number of the moving body from 1.4 to 1.6. The work provides a theoretical basis and the requisite technical support for applications of the gun–track launch system.

**Keywords:** shock wave/vortex; muzzle jet; constrained boundary; reflection; dynamic mesh

## 1. Introduction

A new type of test device called the gun–track launch system has been developed for the non-destructive recovery of high-value warheads by taking full advantage of the characteristics of the testing capabilities of the gun and the rocket track and aiming to satisfy the requirements of a warhead with a muzzle with high kinetic energy [1,2]. This launch system connects the track outside the muzzle, where this constitutes a unique form of motion for the warhead. That is, its state changes from free, high-speed motion in the entire space to constrained, high-speed motion [3,4]. The muzzle jet also exhibits three-dimensional (3D) circumferential asymmetry at this time, which induces vortices of different forms [5,6]. Against this backdrop, the authors of this study raise the problem of the transient development and evolution of the shock wave when the muzzle jet impinges on a constrained moving body. This is important for investigating the complex phenomenon of flow in the muzzle jet with constrained boundaries, which is in turn critical for conducting basic scientific research and solving key technical problems [7,8].

Researchers have used a variety of test platforms for launching projectiles to investigate the performance of the muzzle based on numerical simulations and visual experiments [9,10]. The characteristics of the transient evolution of shock waves due to the impact

of the muzzle jet on a moving object have long been a subject of interest in the area [11,12]. Florio et al. [13] numerically simulated the influence of the area and layout of the exhaust port in the muzzle device on the flow field of the muzzle and considered a configuration with a single opening of different sizes (located near the muzzle). The results showed that increasing the area of the opening reduced the radial range of the muzzle jet, and the radial flow of the lateral opening enhanced the expansion of the gas and its radial diffusion into the environment. Moreover, a large, single-auxiliary opening was found to be more conducive to reducing the pressure of the muzzle.

Schmidt et al. [14] measured the effects of characteristic parameters of the brake of the muzzle on a 20 mm small-caliber gun by using time-accumulated shadow photography. They obtained clear images of the muzzle jet and analyzed its characteristics and mechanism of flow. They reported a strong coupling between the axial flow fields and a weak coupling between the transverse flow fields.

Guo [15,16] used the direct shadow method to carry out visualization experiments on the muzzle jet of a 7.62 mm small-caliber gun and obtained a large number of clear, high-resolution time-series shadow photographs that displayed the characteristics of the typical muzzle jet, including the shock wave, weak compression wave, contact discontinuity, and boundary of the jet, under various conditions. They also discussed the dynamic development of the flow field under different conditions and provided direct experimental comparisons as well as a reference for numerical calculations and related weapons research.

Zhang [17,18] used the shadow method to conduct experimental and numerical simulations on the characteristics of the development of the shock wave of an impinging jet on a moving projectile with a muzzle device. The processes of mutual collision, formation of the lateral jet, and its eventual coupling with the main flow field of the muzzle device were systematically described. The authors observed that a typical, circumferential, symmetrical, multi-level shock wave and a wave system with overlapping discontinuities were formed in the flow field of the muzzle device without a cavity. However, the diverting effect of the open-ended device caused part of the jet to discharge from the side hole in the flow field of the open-ended muzzle device, and its rate of initial axial expansion was greater than that of initial transverse expansion.

Du [19] analyzed the flow field of a 200 mm caliber projectile that was dynamically launched by an electromagnetic track launcher by using the multi-block, structured, overlapping mesh method. The results showed that the dynamic launch of the electromagnetic orbit launcher involved the flow field of a complex shock wave system, with the combined action of a pre-projectile shock wave, a moving spherical shock wave, and a coronal shock wave. The pressure distribution at the midpoint of the head of the projectile was "symmetric," the distribution of the drag coefficient was correlated with the pressure distribution at this point, and the "weak symmetry" of the shock wave at the muzzle first decreased and then increased.

Zhou [20] used a high-pressure gas propulsion device to launch 130 mm caliber projectiles and analyzed the muzzle jet. The results of simulations showed that the jet pushed the projectile forward, and the air flow dispersed and expanded into the space on both sides of the bottom of the projectile to generate vortices. As the projectile accelerated, a Mach disk appeared at its bottom, and the vortices on both sides diffused outward. The Mach disk subsequently moved and shrank, and this led to the appearance of prominent reflected shock waves and slip surfaces.

In sum, researchers have extensively investigated the characteristics of the flow field of a muzzle jet impacting a freely moving body, but few studies have examined the development and mechanism of evolution of the shock and vortex of a muzzle jet impacting a constrained moving body. The innovation of this article is that the authors designed a small-scale test platform with a constrained track to tackle the problem of the evolution of the shock wave of a supersonic jet as it impacts a high-speed moving body. Meanwhile, we established a model with an infinitely long track to simulate the real engineering environment, with the aim of better understanding the complex phenomenon of flow

caused by the special, constrained boundary of the muzzle jet of the gun-track launch system. We compare and analyze the characteristics of the transient evolution of the shock wave by establishing models of the muzzle jet with short and "infinitely" constrained tracks.

The remainder of this paper is structured as follows: Section 2 details the relevant physical model, model of the mesh, boundary conditions, and methods to obtain solutions. Section 3 briefly introduces the small-scale test of the muzzle jet impacting a moving body, including the device and the scheme applied. Section 4 compares and analyzes the results of the experiments and simulations under different constraints on the structure of the muzzle jet, the characteristics of the evolution of the shock wave, the generation and evolution of the vortex, transient changes in pressure at key points, and key parameters of the moving body. Section 5 summarizes the conclusions of this study.

## 2. Physical and Mesh Models

### 2.1. Physical Model

We established two models to realistically analyze the launch environment. Case 1 corresponded to the structure of the small-scale test, while Case 2 involved an infinitely constrained track.

Figure 1a–c show the simulated physical structure of the models considered here. It mainly included a constrained track, a tube, a moving body, and a supporting plate. The diameter of the muzzle was 65 mm, and the constrained track was simplified to a I-shaped, ignoring small grooves. The launcher was cylindrical, 600 mm long, and fitted the structure of the track. We have appropriately simplified the moving body to a cylinder. It was 500 mm above the ground and was 6.0 m long along the direction of motion and 2.5 m long along the other directions.

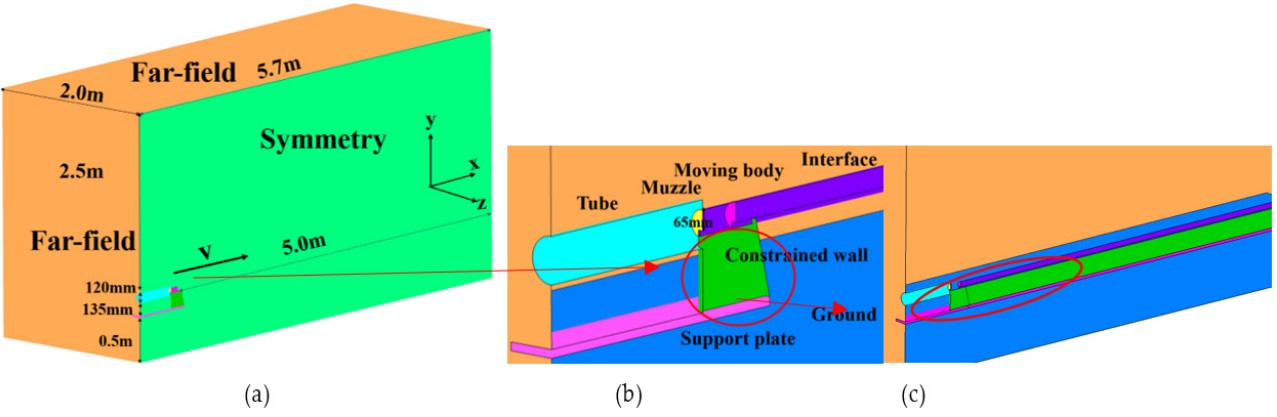

**Figure 1.** Simulated physical structure of the model: (**a**) Physical structure and size of the computational domain. (**b**) Local physical structure of Case 1. (**c**) Local physical structure of Case 2.

### 2.2. Dynamic Mesh Method

The dynamic mesh method was used to handle the changes in the mesh caused by the motion of objects during the calculation. The dynamic mesh algorithm was used to determine the requisite adjustments to nodes in the mesh. Any of three algorithms can be used for this purpose: the layering method, the smoothing method, and the remeshing method. As the object only exhibited translational motion in one direction, we determined that the layering method could best simulate its motion.

It is necessary to set the segmentation and merging factors of the mesh in the layering method. Due to the movement of the body, the mesh behind it moves forward along the X-axis, and this causes the mesh layer near the bottom of the mobile body to stretch, with the length of the edge of the mesh of hs. The mesh layer near the top of the moving body is compressed at the same time, with an edge of length hc. Assuming that the ideal length of the mesh edge is hi, the mesh is divided into two parts when the size of the newly

generated mesh satisfies the condition h > (1+Cs) hi. When its size satisfies h < Cshi, the meshes adjacent to it are merged into one mesh. In this study, we set hi = 2.0 mm, Cs = 0.4, and Cl = 0.2.

### 2.3. Mesh Model

To reduce the cost of the calculations as well as the time needed to perform them, we simulated only half of the model. Figure 2a–c shows a diagram of the mesh model. The entire computational domain was divided into 9.8 million meshes. It is difficult to generate a single, high-quality mesh in many cases when simulating a complex flow field. The mesh needs to be segmented and spliced to facilitate processing. The computational domain was divided into domains with static and dynamic meshes, and their interface was used for numerical exchange to this end.

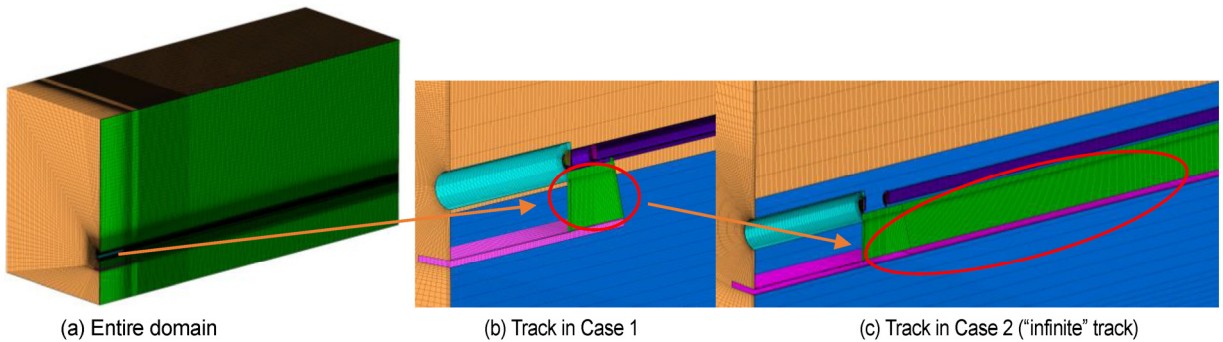

(a) Entire domain      (b) Track in Case 1      (c) Track in Case 2 ("infinite" track)

**Figure 2.** Schematic diagram of the mesh model.

### 2.4. Boundary Conditions and Solution Methods

We simulated the model of a supersonic jet impinging on a high-speed moving body by using the commercial software Fluent. The domains of calculation were set as the boundary conditions at the outlet of pressure, while the tube and track were assumed to be non-sliding insulation walls. The moving body was assumed to adhere to the boundary condition of the moving wall and was controlled by a user-defined function (UDF) program with six degrees of freedom (six-DOF). The initial velocity was set to 500 m/s, according to the test data. The boundary condition for the muzzle was the boundary at the inlet of pressure, which was 45 MPa according to the test. According to the after-effect model, the pressure of the muzzle changed with time as $p = 45.0 \times 10^6 e^{-24.886}t$. The environmental pressure was 101,325 Pa, and the temperature was 300 K.

We processed the motion in the given block-based division of the mesh based on the finite volume method in combination with the structural dynamic method. The Navier–Stokes (N–S) equation solver based on density correction and the K-$\varepsilon$ realizable turbulence model were used. The inviscid convective flux was split by using the Roe-FDS scheme, and the implicit scheme was used with the time marching method to accelerate the convergence of the numerical calculations. The equation of flow control was discretized by using the second-order upwind scheme, and the material was assumed to be an ideal gas.

### 2.5. Mesh Independence Verification

To verify the independence of the mesh, we designed three sets of meshes, namely 2.2 million, 6.0 million, and 9.8 million. They were respectively defined as Coarse mesh, Fine mesh, and Dense mesh. Regarding the variation in Ma of the moving body in Case 1, Figure 3 demonstrates the results of numerical simulations of the different mesh models considered. From the figure, the results obtained by the Fine mesh and Dense mesh were very close, with an error of less than 1%. On the contrary, there were significant differences in the results obtained by Coarse mesh model. Therefore, it can be concluded that when using Fine mesh model, the simulation results were independent of the size of the mesh. Therefore, we used Fine mesh model in our calculations.

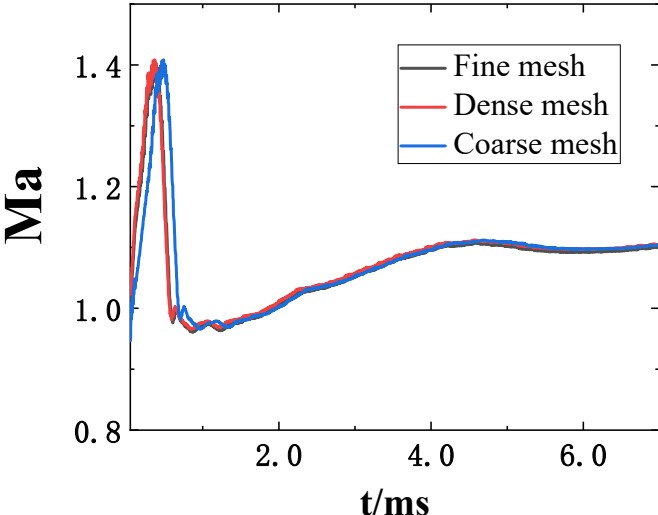

**Figure 3.** Comparison of Ma of the moving body to verify mesh independence.

## 3. Analysis of Test Results

### 3.1. Test Equipment

We developed a small-scale test of the muzzle jet as it impacted a high-speed moving body in the presence of a constrained track. The test device shown in Figure 4 was a 65 mm caliber gun with a constrained track installed at its muzzle to construct a special constrained boundary.

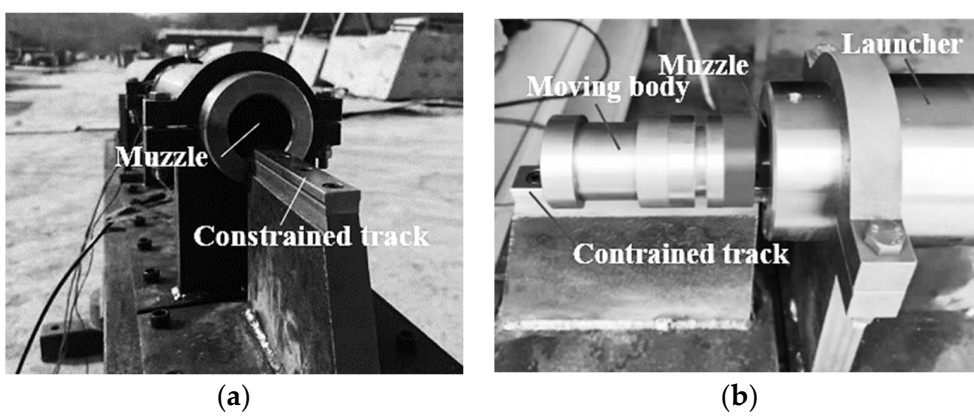

<div align="center">(a)       (b)</div>

**Figure 4.** Photographs of the test mode: (**a**) Photograph of the constrained boundary. (**b**) Photograph of the moving body.

### 3.2. Test Plan

Figure 5 illustrates the test plan. The test platform was the main generator of the jet, and the moving body was pushed out of the muzzle at a speed of about 500 m/s. The muzzle was externally connected to a constrained track, and the body moved along it after having been discharged. A piezoelectric dynamic pressure sensor, manufactured by the Kistler company, was used to measure the overpressure signals. All pressure signals were collected using the DEWE2-A13 transient data recorder that was manufactured by DEWETRON Industrial Measurement System (Grambach, Austria). The formation and development of the overpressure field were photographed by using a high-speed photon camera at a frame rate of 5000 fps.

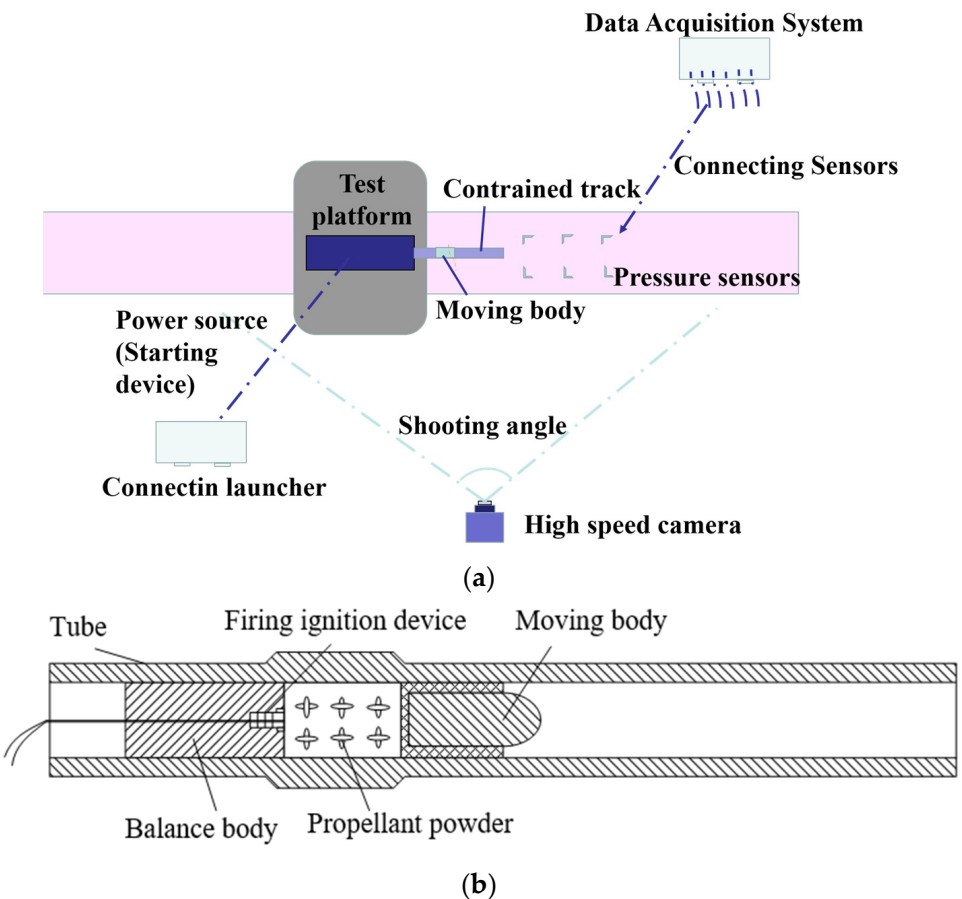

**Figure 5.** Schematic diagram of the test. (**a**) Schematic diagram of the test plan, (**b**) Schematic diagram of the internal loading of the launch device.

### 3.3. Test Results

During the experiment, we triggered the ignition system to ignite the gunpowder inside the chamber, generating high-temperature and high-pressure gas that propelled the moving body forward at high speed. Meanwhile, we applied the high-speed camera to capture the development posture of the muzzle jet and the movement posture of the moving body to determine the operating speed of the moving body. At the same time, the pressure sensor is triggered to monitor the pressure at key points. Figure 6 shows the test procedure as captured by the high-speed camera. The moving object had just appeared from the muzzle in Figure 6a, marking the beginning of the simulation, and the muzzle jet had just begun to form at this time. Figure 6b shows that the moving body flew out of the muzzle, and high-pressure and high-temperature gas and smoke were ejected from it and impacted the moving body. With the movement of the high-speed projectile and the development of the jet, the muzzle jet came into contact with the ground and generated a reflected wave front in the opposite direction. The new wave front continued to develop upward and is clearly displayed on the background plate. It significantly interfered with the development of the muzzle jet. As the latter developed, the reflected wave formed a local high-pressure area under the muzzle. The numerical simulations provided below account for this phenomenon in detail. Figure 6c shows that the moving object gradually flew out of the muzzle over time, and the muzzle jet finally disappeared as the gunpowder was exhausted.

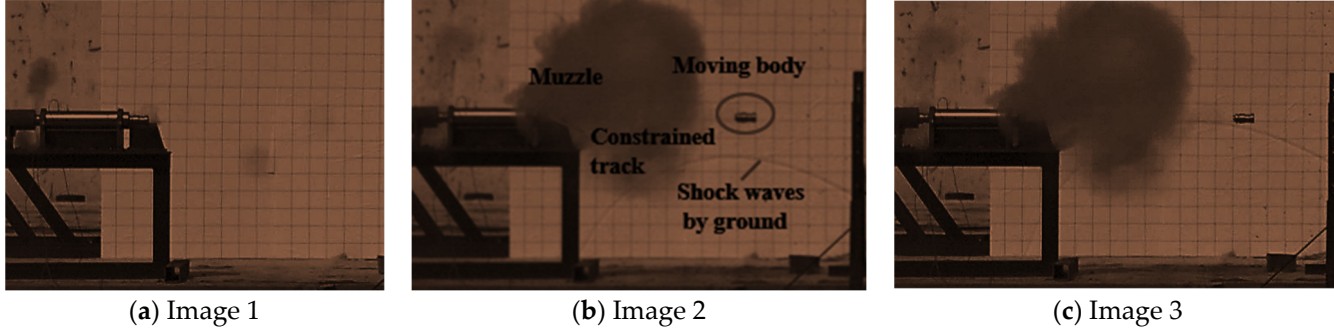

(**a**) Image 1　　　　　　　　　　(**b**) Image 2　　　　　　　　　　(**c**) Image 3

**Figure 6.** Representative visualized results of the test.

Changes in the pressure at key points were also monitored during the test. Figure 7 compares the curves of pressure versus time between the test and the simulation at the points (1500, 30, 10) mm, and it is clear that they were consistent. Therefore, the method of simulation used in this paper could accurately simulate the characteristics of the muzzle jet.

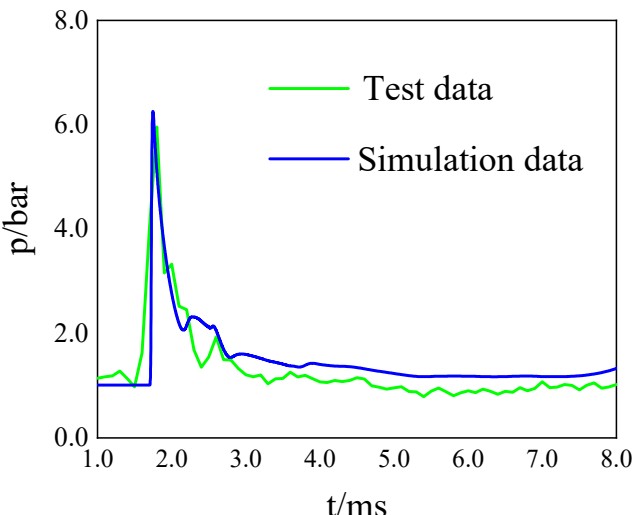

**Figure 7.** Comparison of the changes in pressure over time at a key point between the test and the simulation.

## 4. Results of Numerical Simulations

### 4.1. Muzzle Jet

The muzzle jet poses a series of aerodynamic problems, among which shock/shock interference, a strong shock, and a strong vortex are the main phenomena of flow in the flow field. Compared with that in the traditional muzzle jet, the moving body moves forward on the track as driven by the jet of gas in the muzzle jet, which is affected by a constrained track, and this limits the state of development of the muzzle jet in free space. Therefore, it exhibits a high degree of 3D circumferential asymmetry and generates various kinds of vortices.

Figures 8 and 9 show the contours of pressure of the muzzle jet on the plane of symmetry in the two cases. Figure 8a,b show that the moving body had just moved out of the muzzle to form the preliminary muzzle jet. However, it developed inadequately, and its range of influence was small. In addition to being hindered by the constrained track, it pushed the surrounding air to form a "semi-spherical" shock wave, in addition to a shock wave that was simultaneously formed in front of the moving body. The wave front of the shock wave was relatively regular and spherical, but it was truncated on the track.

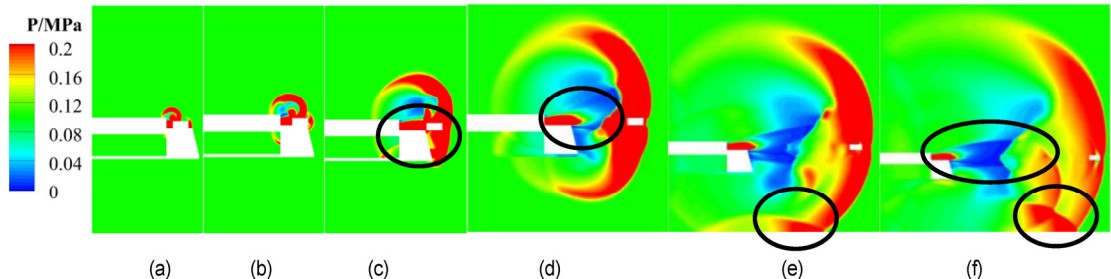

**Figure 8.** Contours of pressure of the muzzle jet on the plane of symmetry over time in Case 1: (**a**) t = 0.1 ms; (**b**) t = 0.2 ms; (**c**) t = 0.5 ms; (**d**) t = 1.0 ms; (**e**) t = 2.0 ms; (**f**) t = 3.0 ms.

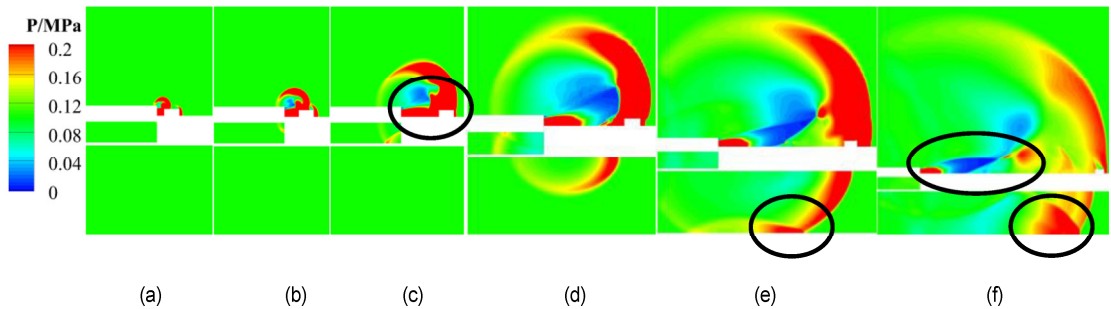

**Figure 9.** Contours of pressure of the muzzle jet on the plane of symmetry over time in Case 2: (**a**) t = 0.1 ms; (**b**) t = 0.2 ms; (**c**) t = 0.5 ms; (**d**) t = 1.0 ms; (**e**) t = 2.0 ms; (**f**) t = 3.0 ms.

Figure 8c,d show that the flow field of the muzzle gradually developed, and its asymmetry became more prominent. Under interference by the constrained track, the support plate, and the test platform, the jet tilted prominently toward the area above the muzzle. Although the track used in the test was not very long, the development of the shock wave was dependent on the past state of the system. The initial disturbance distorted the shock wave of the muzzle so that it changed from being circumferentially symmetric to circumferentially asymmetric. Figure 8e,f show that because the test platform was 500 mm above the ground, the shock wave of the muzzle was emitted when it impacted the ground, which caused it to propagate upward and form a new wave front. A complete wave system consisting of an asymmetric coronal shock wave, a reflected shock wave, and a multi-level shock wave composed of the Mach disk and overlapping discontinuities was formed around the muzzle at this time.

A comparison of the two cases shows that the muzzle jet, when influenced by varyingly constrained tracks, exhibited certain differences. The states of the muzzle jet shown in Figure 8a,b, and Figure 9a,b were completely consistent because they differed in time by only a few tenths of a millisecond, and the moving body was still within the range of the short track. The shock wave moved out of track and gradually developed downward, as shown in Figure 8c,d, while Figure 9c,d show that the shock wave was blocked by the "infinitely" constrained track in Case 2 and could not develop smoothly to the lower area. Figure 9f shows that the difference between the cases gradually became prominent. In Case 2, the wave reflected by the ground encountered a long track and was reflected once again. The reflected wave propagated downward to form a second area of local high pressure. This phenomenon did not occur in Case 1 and is discussed in detail below.

### 4.2. Evolution of Shock Waves and Vortices

The pressure induced by the reflection of the shock wave in the muzzle from the wall was consistently higher than that of the incident shock wave if the angle between the latter and the wall was non-zero. $p_0$, $\rho_0$, and $\mu_0 = 0$ represent the pressure, density, and velocity of the particles of gas at the front of the incident shock wave, respectively, and

$p_1$, $\rho_1$, $D_1$, and $\mu_1$, and $p_2$, $\rho_2$, $D_2$, and $\mu_2$ represent the pressure, density, and velocities of the wave and particles of the gas behind the surfaces of the incident and the reflected shock waves, respectively. The relationship given below was found to hold during the collision that led to the formation of an unsteady and strong shock wave system. The normal reflection-induced excessive pressure of the strong shock wave can be expressed as follows [21]:

$$p_2 = p_1 + \frac{2k\varphi_1(p_1 - p_0)}{(k-1)p_1 + (k+1)p_0} \tag{1}$$

where $k = 1.4$.

The normal reflection-induced pressure was high because the airflow behind the reflected shock wave was abruptly slowed down by the surface of the ground. Figure 10a shows a sketch of the normal reflection of the shock wave, where the subscripts 0–2 represent the corresponding states of gas. As the ground acted as a rigid body, a zone of stagnation was formed between the wall and the reflected shock wave.

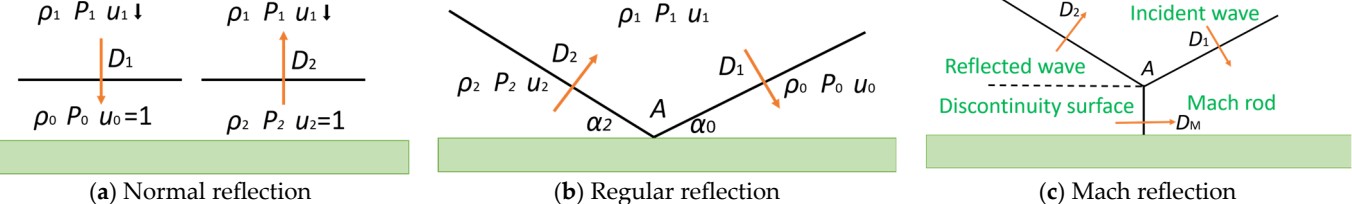

| (**a**) Normal reflection | (**b**) Regular reflection | (**c**) Mach reflection |

**Figure 10.** Schematic diagram of reflection of the shock wave.

A sketch of the regular reflection of the shock wave is shown in Figure 10b. Two kinds of reflections were observed: regular and Mach reflections, depending on the incident angle. The relationship between the regular oblique reflection and the pressure was as follows:

$$\frac{p_2}{p_1} = \frac{7}{6}M_1^2\sin^2\varphi_1 - \frac{1}{6} \tag{2}$$

where $\varphi_1$ is the jumping angle of densification.

When Mach reflection occurred, one wave propagated through the gas near the surface, but two waves (the incident and the reflected waves) propagated farther away through the gas.

Figures 11 and 12 show changes in the development of the vortex in the muzzle jet on the plane of symmetry over time. Under interference by the constrained track, support plate, and test platform, the shapes of the vortices in areas above and below the muzzle differed in Case 1. Both vortices were located in the low-pressure area. The streamline of the vortex in the area above the muzzle pointed outward from its center, and the spiral point was stable and difficult to dissipate. Figure 11b–d show that the development of the vortices in the flow field changed significantly. The vortex above continued to develop steadily, with its streamline pointing from its center to the outside. Its spiral point was stable and difficult to dissipate. The streamline of the vortex in the area below the muzzle pointed to its center from the outside, owing to continuous interference by the wave reflected from the ground. Its spiral point was unstable and easily dissipated.

Figure 12 shows the significant difference between the cases considered. The vortex in the upper region in Case 2 was similar to that in Case 1. It was located in a low-pressure area, and its streamline maintained its direction from its center to the outside, with a stable spiral point that was not easy to dissipate. Interestingly, however, no vortex was found in the lower region. The shock wave of the muzzle was emitted when it came into contact with the ground, and the reflected wave propagated upward. Part of it moved forward while the other part moved backward, and both formed a high-pressure area. The wave reflected by the ground was reflected once again when it met the track and subsequently

developed downward to form a local high-pressure area. Figure 12e–g shows that pressures at the points of reflection were relatively high and the difference in pressure between the points was small. With the continuous development of the flow field, multiple reflections were formed between the ground and the constrained track. This state failed to yield a stable, low-pressure area such that no vortex was formed.

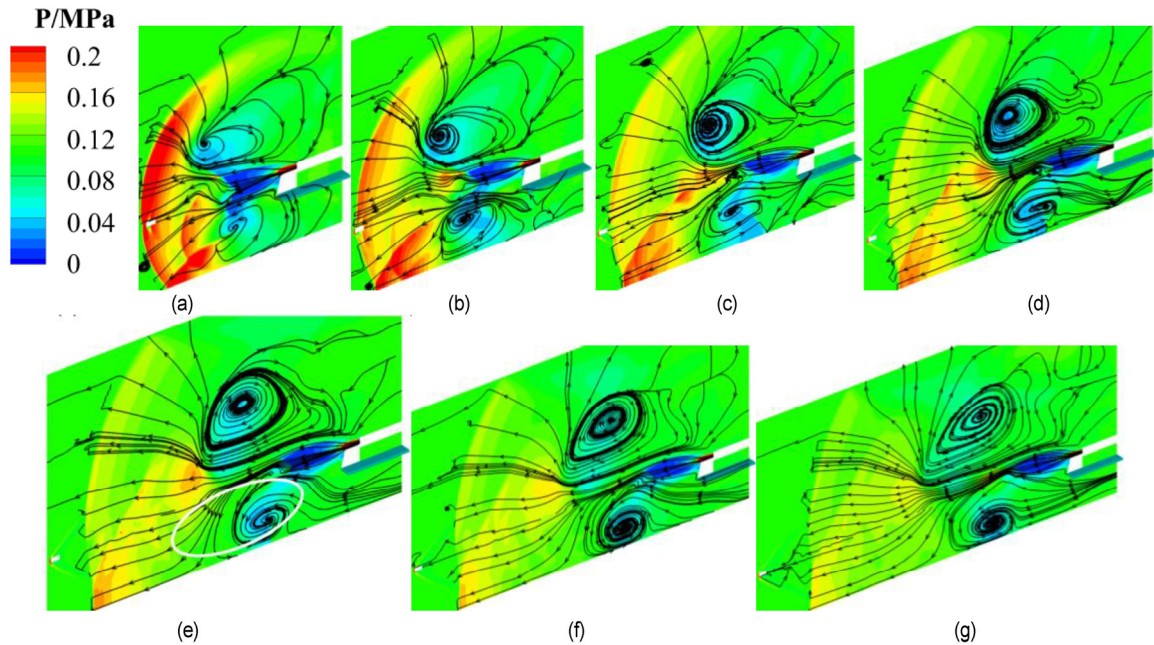

**Figure 11.** Evolution of vortices in a symmetric muzzle jet over time in Case 1: (**a**) t = 3.0 ms; (**b**) t = 4.0 ms; (**c**) t = 5.0 ms; (**d**) t = 6.0 ms; (**e**) t = 7.0 ms; (**f**) t = 8.0 ms; (**g**) t = 9.0 ms.

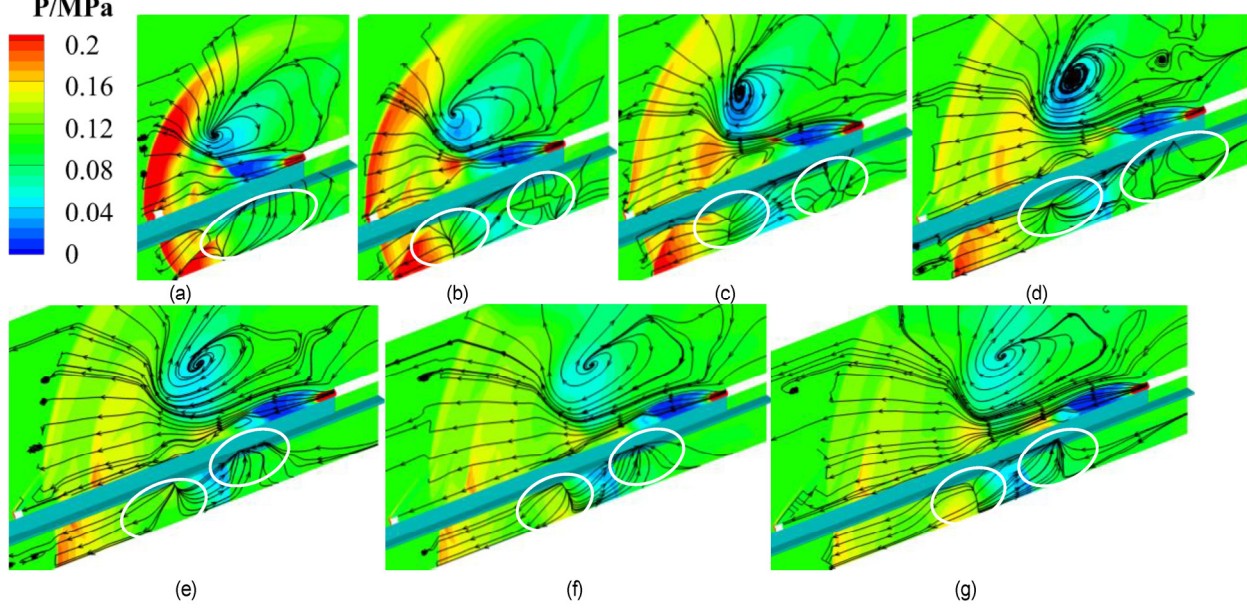

**Figure 12.** Evolution of vortices in a symmetric muzzle jet over time in Case 2: (**a**) t = 3.0 ms; (**b**) t = 4.0 ms; (**c**) t = 5.0 ms; (**d**) t = 6.0 ms; (**e**) t = 7.0 ms; (**f**) t = 8.0 ms; (**g**) t = 9.0 ms.

Figure 13 shows a comparison of the contours of the development and evolution of the shock waves in the area below the muzzle in the two cases. It is clear from Figure 13a that in Case 1, the shock wave of the muzzle was reflected from the ground and formed a local area of high pressure and another shock wave. The shape of the shock wave in

Case 2 was different from that in Case 1. The wave reflected from the ground was reflected once again when it encountered the track and bent downward. As shown in Figure 13b, the shock wave continues to evolve. In Case 1, as the reflection angle increased, the point of incidence was gradually lifted off the ground and formed a Mach rod. In Case 2, the shape of the second shock wave, formed by contact with the track, became increasingly prominent, and there was no significant low-pressure area in the region. This also explains why there was no vortex in Case 2. Figure 13c shows that in Case 1, the shock wave evolved continuously, and another shock wave was generated at "Position 2." The Mach rod was significantly higher than that in Case 2. The launch point on the track in the latter case shifted forward, and a prominent shock wave was generated at Position 2. Figure 13d,e show that the height of the Mach rod in Case 1 gradually increased and the reflected shock wave gradually diminished. The height of the Mach rod gradually increased in Case 2 as well, while the reflected shock gradually diminished, and the shock had a prominent shape at Position 2. On the whole, the shock wave in Case 1 was launched from the ground and formed another shock wave, while the shock wave in Case 2 was reflected multiple times between the ground and the constrained track to form a multi-channel shock wave.

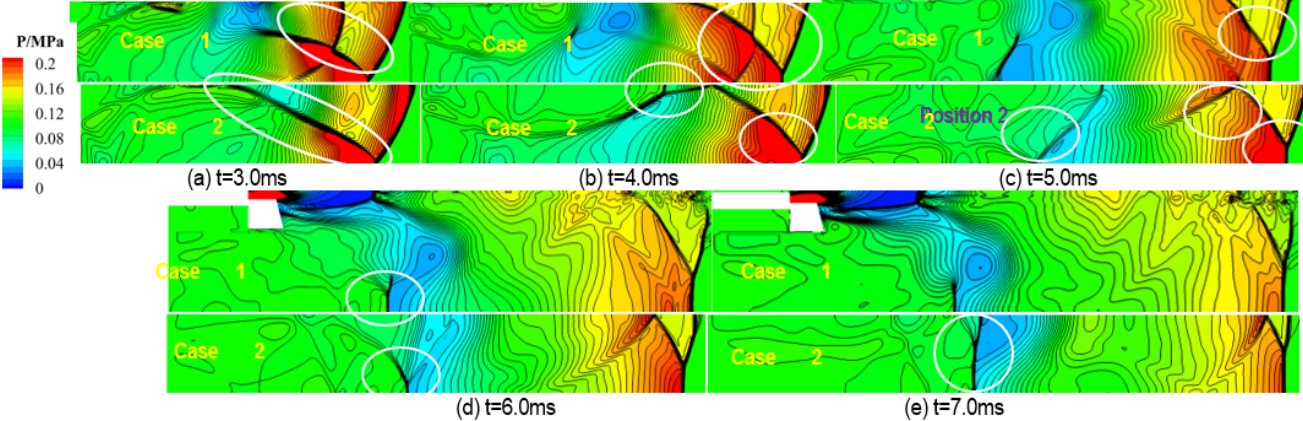

**Figure 13.** Comparative contours of shock waves in the area below the muzzle in the two cases: (**a**) t = 3.0 ms; (**b**) t = 4.0 ms; (**c**) t = 5.0 ms; (**d**) t = 6.0 ms; (**e**) t = 7.0 ms.

*4.3. Comparison of Pressure*

To further illustrate the impact of different constrained boundaries on the shock wave and the vortex, we monitored the changes in pressure at several key points over time for comparative analysis. Figure 14 shows a schematic diagram of the locations of the monitoring points, and Table 1 lists their locations.

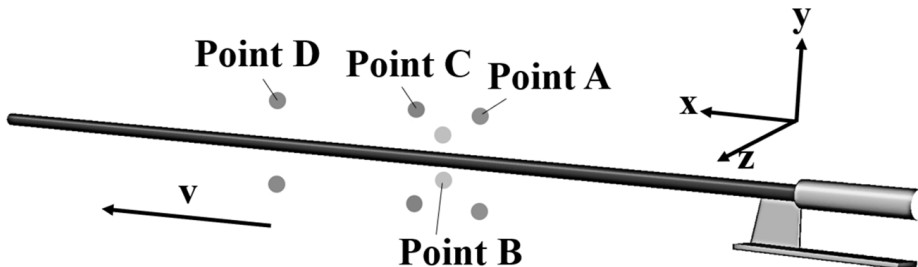

**Figure 14.** Schematic diagram of the locations of the monitoring points.

**Table 1.** Locations of the monitoring points on the track.

| Point | Location (x, y, z)/mm |
|---|---|
| A-above | (1600, 60, 200) |
| A-below | (1600, 60, −200) |
| B-above | (1800, 35, 100) |
| B-below | (1800, 35, −100) |
| C-above | (2000, 60, 200) |
| C-below | (2000, 60, −200) |
| D-above | (3000, 0, 200) |
| D-below | (3000, 0, −200) |

Figures 15a–d and 16a–d compare the changes in pressure over time at the key points in Case 1. They show that the muzzle jet exhibited an upward trend, and the monitoring points above the muzzle were disturbed earlier than those below it.

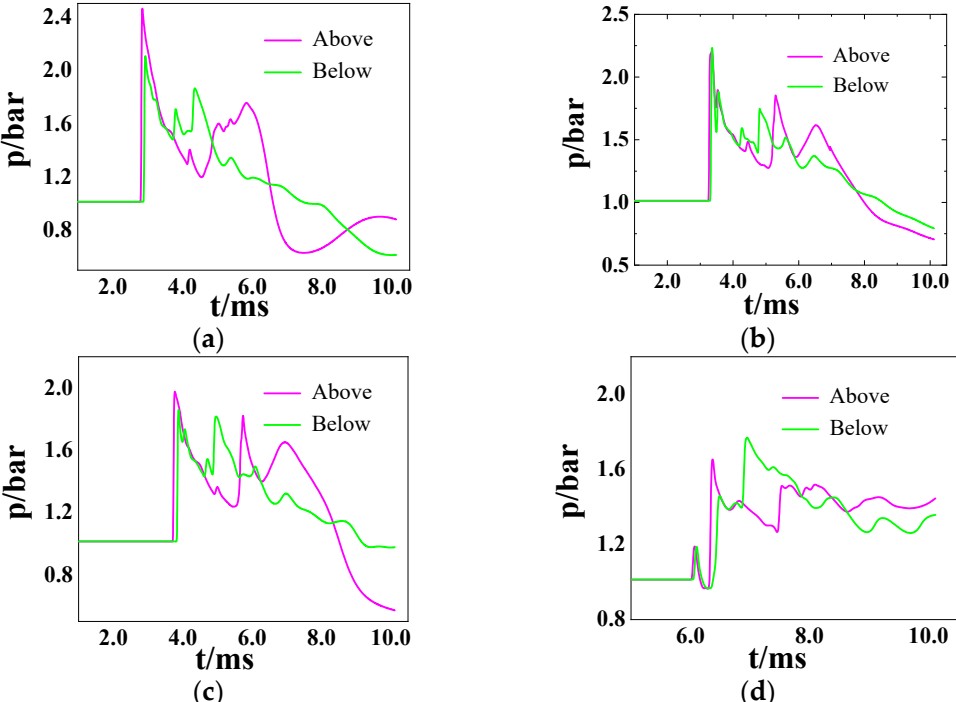

**Figure 15.** Comparisons of pressure at each monitoring point in Case 1 overtime: (**a**) Point A. (**b**) Point B. (**c**) Point C. (**d**) Point D.

Figure 15a shows that the body moved at a velocity of 500 m/s, and the shock wave moved to this monitoring point in 3 ms to increase the pressure. The maximum pressure at A-above should have been slightly higher than that at A-below. The pressure then decreased and slightly rose at about t = 6.0 ms. The pressure at A-below was supplemented by the reflected waves, because of which the fluctuations in it were relatively small. The shock wave gradually moved forward, and the pressure became stable. Figure 15b shows that the shock wave moved to Point B at t = 4.0 ms. Because Point B was close to the centerline of the jet, the difference in pressure at its point of symmetry was relatively small compared with those at the other monitoring points. The maximum pressures at B-above and B-below were close to each other, and the overall changes in pressure were similar. Figure 15c shows that at t = 4.0 ms, the shock wave moved to Point C, and the change in pressure at C-below was smoother than that at C-above. The change in pressure was small in the area below the muzzle because it was supplemented by the reflected wave, while the pressure in the area above the muzzle was not supplemented. Due to the distance between

Point C and the centerline of the jet, the disturbance caused by the shock wave continued over time. Figure 15d shows that the maximum pressure at D-below was greater than that at D-above. Because Point D was far from the muzzle and the muzzle jet developed more fully, it was significantly influenced by the ground, and this led to a wide range of high pressure in the lower part such that the maximum pressure at D-below was higher than that at D-above.

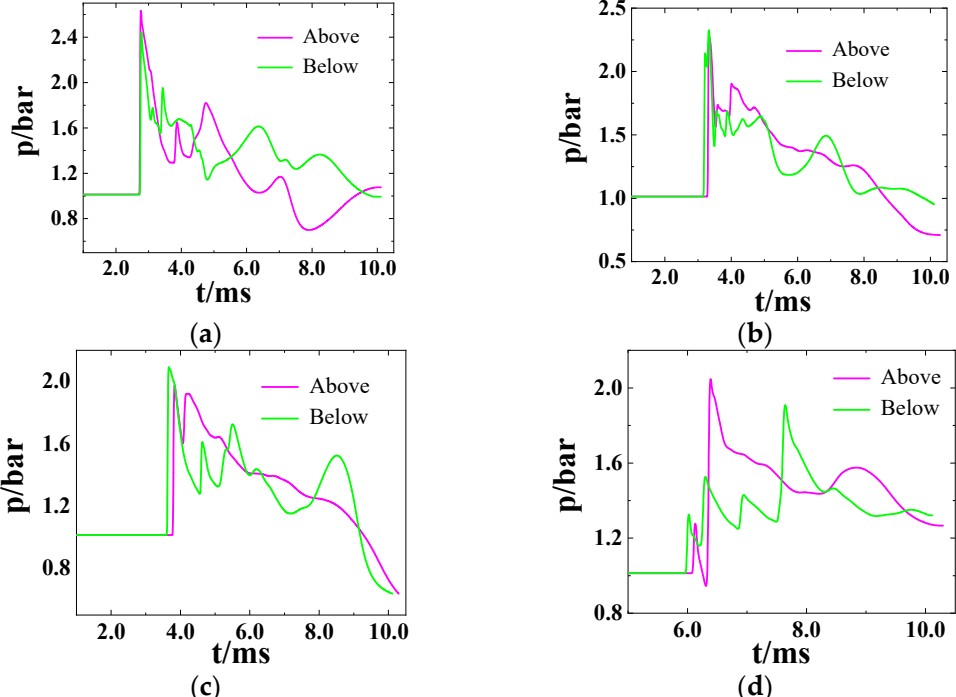

**Figure 16.** Comparisons of pressure at each monitoring point in Case 2 overtime: (**a**) Point A. (**b**) Point B. (**c**) Point C. (**d**) Point D.

However, the changes in pressure at the same monitoring points were significantly different in Case 2 from those in Case 1, and fluctuations in it were prominent. This also shows that the shock wave was constantly reflected between the ground and the constrained track. The disturbance due to the shock wave influenced the pressure at Point A at almost the same time in both cases, which indicates that the speed of propagation of the shock wave in the lower area of Case 2 was higher than that in Case 1. As for the pressure at Point A in Case 1, the change in the amplitude of pressure at A-below was smaller than that at A-above, but the overall fluctuations in it were prominent. It is clear from Figure 16b that the disturbance in pressure at B-below occurred earlier than that at B-above, where this is different from the situation in Case 1. Figure 16d shows that the shock wave of the muzzle had fully developed by 6.0 ms. Due to interference by multiple reflected waves, the shape of the area under the constraining rail was completely different from that of the shock wave above it. Therefore, the changes in pressure in these areas were also significantly different.

Figure 17 shows a comparison between the cases at the same monitoring point. As shown in the figure, the speed of propagation of the shock wave in Case 2 was higher than that in Case 1, which indicates that the infinitely constrained track in Case 2 accelerated its evolution. The maximum pressure in Case 2 was significantly higher than that in Case 1, which shows that the infinitely constrained track in Case 2 increased the maximum pressure of the shock wave. This is because during the rapid expansion process of the muzzle jet, the long track further limits the expansion of the shock wave, so the attenuation and decay of the shock wave will be slower, resulting in faster velocity and higher pressure. In addition, the trend of changes in pressure over time varied at all four monitoring points, which

shows that the constrained track not only influenced the evolution of shock waves and vortices in the area below the muzzle but also had an impact on the area above the muzzle.

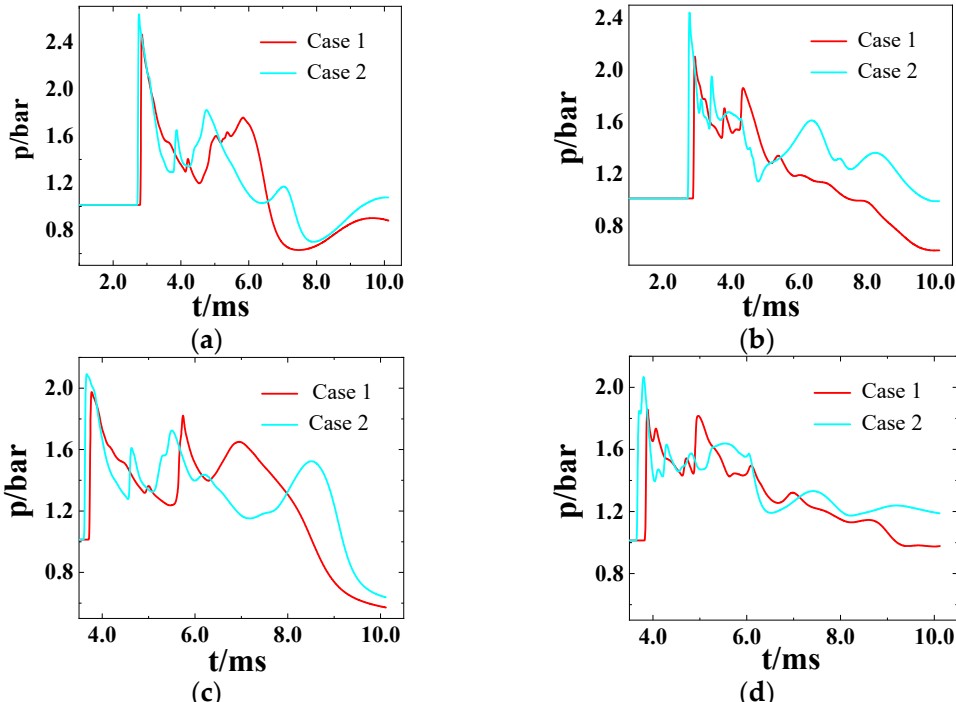

**Figure 17.** Comparisons of pressure at the monitoring points over time in Case 1 and Case 2: (**a**) Point A-above. (**b**) Point A-below. (**c**) Point C-above. (**d**) Point C-below.

### 4.4. Key Parameters of Moving Bodies

Figure 18 shows the drag forces, Mach numbers, and total thrust of the moving body in both cases. It is clear from Figure 18a that the resistance of the moving body in Case 1 was a bit larger than that in Case 2. The two cases have similar fluctuations. This also shows that the speed of the moving body in Case 2 was lower than that in Case 1. The change in the Mach number and total pressure of the moving body in Figure 18b,c verifies this. As shown in Figure 18b, the velocity of the moving body increased close to the muzzle due to the thrust of the muzzle jet. As the body moved away from the muzzle, its propulsion rapidly weakened, such that its Mach number gradually decreased under the action of resistance. The Mach number of the moving body in Case 2 was significantly higher than that in Case 1, indicating that the track restricted the development of the shock wave of the muzzle and caused it to become concentrated on the track. In Figure 18c, the maximum thrust at the bottom of the moving body is 45 Mpa, which then rapidly decreased as it moved forward. At the beginning, the trend of the two cases was basically the same because both cases had a track at the muzzle. Next, the thrust in Case 1 continued to decrease, while the thrust in Case 2 had an upward process. This is because the long rrack constrains the development of the muzzle jet, preventing it from spreading smoothly around but rather concentrating more on the orbit, which acts on the bottom of the moving body. This is also why the Mach number of the moving body in Case 2 is higher, as it receives more thrust.

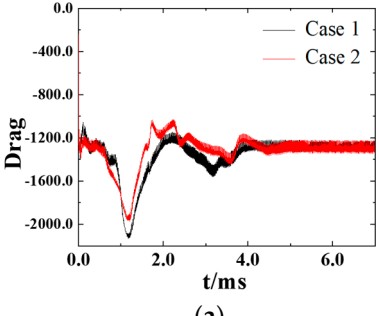 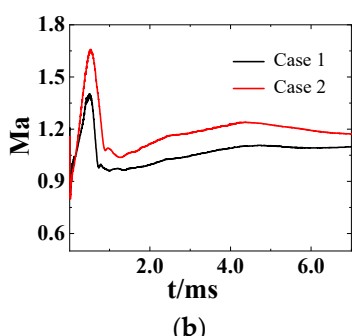 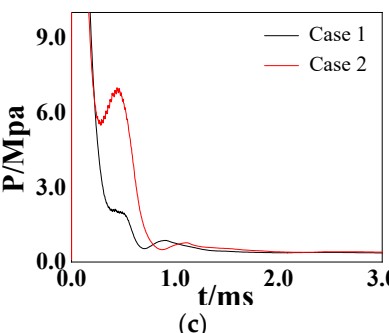

(**a**) (**b**) (**c**)

**Figure 18.** Drag forces, Mach numbers, and total thrust of the moving body over time in Cases 1 and 2: (**a**) Drag. (**b**) Mach number. (**c**) Total thrust.

## 5. Conclusions

In this study, the authors established an experimental platform and models of simulation by using the dynamic mesh method to investigate the characteristics of the evolution of shock waves and vortices in the case of a muzzle jet impinging on a moving body in the presence of varyingly constrained tracks. The conclusions are as follows:

(1) The shock wave of the spherical muzzle, which should have been circumferentially symmetrical, was distorted due to the presence of the constrained track, and the entire structure was inclined to the area above the muzzle. Because the shock wave was blocked by the track, transverse air flow was generated, which led to the formation of vortices. The characteristics of the evolution of the vortices were unique and thus different from those of the traditional model of the muzzle jet.

(2) Because the test platform was close to the ground, the shock wave of the muzzle was reflected from it, which enriched the characteristics of the evolution of the shock wave and the vortex in the asymmetric muzzle jet. This led to the formation of a Mach rod and a vortex in the area above the ground in Case 1. However, in Case 2, the shock wave was reflected multiple times between the ground and the constrained rail without forming a stable low-pressure area or a vortex.

(3) The constrained track influenced the development and evolution of the muzzle jet, where this was directly reflected by the difference in pressure and the velocity of the shock at the monitoring points between the cases considered. At the same time, the Mach number and total thrust of the moving body in Case 2 were higher than those in Case 1.

**Author Contributions:** Conceptualization, Z.L.; methodology, Z.L.; software, Z.L.; validation, Z.L., H.W.; formal analysis, Z.L.; investigation, Z.L.; resources, Z.L., H.W.; data curation. Z.L.; writing—original draft preparation, Z.L.; writing—review and editing, Z.L., H.W; visualization, Z.L.; supervision, Z.L.; project administration, Z.L.; funding acquisition, Z.L. All authors have read and agreed to the published version of the manuscript.

**Funding:** This research received no external funding.

**Data Availability Statement:** All data and models that support the findings of this study are available from the corresponding author upon reasonable request. All data and models generated or used during this study appear in the submitted article.

**Acknowledgments:** Thank you to Hexia Huang (Nanjing University of Aeronautics and Astronautics) for his opinions, guidance, and contributions to this article.

**Conflicts of Interest:** The authors declare that they have no conflict of interest.

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
