# Peer review of "Evolution of Shock Waves during Muzzle Jet Impinging Moving Bodies under Different Constrained Boundaries"

_aerospace, doi:10.3390/aerospace10110908_

Round 1

Reviewer 1 Report

Title:  Evolution of Shock Waves During Muzzle Jet Impinging Moving Bodies under Different Constrained Boundaries

1. In abstract obtained results have to represent in term of numerical value. Novelty has to clearly define after introduction.

2. In dynamic mesh method optimum mesh size has to define clearly.

3. Experimental procedure and data reduction method have to clearly define.

4. The resolution of the figure 5, 6, 14, 15, 16, 17, 18 have to improve.

5. More recent and relevant references are required. Two references have been included below:

DOI: 10.12989/aas.2023.10.3.203

**********

It is ok.

Author Response

Dear Editor

Due to the inability to display images on the page, please find my response in the attachment.

Yours sincerely,

Zijie Li

Reviewer 2 Report

The paper reports some numerical simulations and experimental measurements for the investigation of the characteristics  of shock waves and vortices developing from a muzzle jet impinging on a moving body in the presence of tracks. A dynamic mesh method was employed to account for the changes in the mesh caused by the moving body in the numerical simulations. Some comments are included below.

It is difficult to identify the novelty of this paper because the authors mostly focus on describing the methodologies and results, but very little explanation is provided about why the results appear this way. Just as an example (among many other), on page 12 the authors mention that “Fig. 17 shows a comparison between the cases at the same monitoring point. The speed of propagation of the shock wave in Case 2 was higher than that in Case 1, which indicates that the infinitely constrained track in Case 2 accelerated its evolution”, but no explanation about why?

Most figures must be made large, and more resolution must be considered ib images to increase the readability (see for example, figure 6).

Include references for the equations (for example, equation (1)).

Include a reference for the commercial solver Fluent, and provide the version number of the code.

Is the name of section 3.2 correct? (it’s the same as section 3.1)

Show a schematic of the experimental rig.

The English is generally good. Few typos have been spotted in the text, so I recommend the authors to thoroughly read proof the manuscript.

Author Response

(The authors gave the same response as above.)

Reviewer 3 Report

The paper describes a numerical simulation of complex shock wave interactions. Although tests (including pressure measurements (Sec 3.1/3.2) were performed and briefly described no comparison with measured data (except Fig 7) was shown. Consequently, it is rather difficult to assess the accuracy of the simulation. However, the physical description of the numerical results are reasonable. The paper could be better if the comparison measurement vs calculation had been added and discussed.

The authors should discuss following topics.

1.    How did the authors verify that the layer technique used to take into account the mesh movement is numerical non-reflecting?

2.    The simulation data of the time-depended pressure (Fig 7) looks partial reasonable only (until the first peak). But there are two questions:

a.     Why the pressure curve does not reach the ambient / initial (t=1.0) pressure level later in time?

b.     Why pressure curve does show a pressure increate at t=8.0? Are there any reflected waves disturbing the pressure field?

3.    Checking the function p(t) in Sec 2.4 – it does not fit to the pressure plots Fig 7, 15-18 and 8,9. Or I ask the authors to describe the boundary conditions of the muzzle region more precisely. Doesn’t the authors experiences numerical instabilities between the muzzle with a pressure level of 45MPa and the remaining area (ambient) with 0.1MPa? It is more the 2 order of magnitude difference. Did the authors check the fluid properties in the vicinity of the muzzle (not the muzzle itself)? How does the shock wave structure look in that region?

4.    What is the typical numerical time step size for the simulation? How was checked that the solution is converged? What was the total physical time simulated? What was the muzzle pressure at the physical end time of the simulation?

Some recommendations:

1.    Text in Figure 1 is too small to read/print, especially in Fig 1 b) and c)

2.    adding an arrow showing the movement direction

3.    showing the X-axis orientation (as described in Sec 2.2)

4.    A short notice about the mesh sizes including an assessment about the mesh size dependence of the simulation is helpful for the reader

5.    increasing the contrasts in Fig 6  (photographs) to visualize the shock waves better

6.    Checking the function p(t) in Sec 2.4 – it does not fit to the pressure plots Fig 7, 15-18 and 8,9. Or I ask the authors to describe the boundary conditions of the muzzle more precisely. Doesn’t the authors experiences numerical instabilities between the muzzle region with a pressure level of 45MPa and the remaining area (ambient) with 0.1MPa? It is more the 2 order of magnitude difference. Did the authors check the fluid properties in the vicinity of the muzzle (not the muzzle itself)?

written clearly with some typos

Author Response

(The authors gave the same response as above.)

Round 2

Reviewer 1 Report

Rebuttals and revised manuscript should be same. 

Detailed opinion #5:

Page no. 6 in rebuttals. 

Author Response

Dear Editor,

Attachment please find the modify reply, thank you.

Best regards,

Zijie Li

Reviewer 3 Report

Dear authors, 

according the explaination you gave about the Interface layer technique I see you didn't check the numerics concering numerical reflections. You are lucky that Fluent did a great job. You should ckeck it in case you are using other software (keyword non-refecting boundary conditions).

Author Response

(The authors gave the same response as above.)
